# Assessing health governance across countries: a scoping review protocol on indices and assessment tools applied globally

Aidan Huang,[1,2] Yuling Lin [ORCID],[3] Liyuan Zhang,[4] Jingwen Dong,[5] Qiwei He,[1,2] Kun Tang [ORCID] [1,2]

¹Vanke School of Public Health, Tsinghua University, Beijing, China
²Institute for Healthy China, Tsinghua University, Beijing, China
³Global Studies Institute, University of Geneva, Geneva, Switzerland
⁴Department of History and Philosophy of Science, University of Cambridge, Cambridge, UK
⁵School of Public Health, Shanghai Jiao Tong University, Shanghai, China

**Correspondence to**
Dr Kun Tang;
tangk@mail.tsinghua.edu.cn

## ABSTRACT

**Introduction** Most global health indices or assessment tools focus on health outcomes rather than governance, and they have been developed primarily from the perspective of high-income countries. To benchmark global health governance for equity and solidarity, it becomes necessary to reflect on the current state of indices or assessment tools evaluating health governance across countries. This scoping review aims to review the existing multicountry indices and assessment tools applied globally with measurable indicators assessing health governance; summarise their differences and commonalities; identify the lessons learnt through analysis of their advantages and gaps; and evaluate the feasibility and necessity to establish a new index or consensus framework for assessing global health governance.

**Methods and analysis** This scoping review protocol follows Arksey and O'Malley's methodological framework, the Joanna Briggs Institute guidelines and the Preferred Reporting Items for Systematic Reviews and Meta-analyses methodology for scoping reviews. Key information sources will be bibliographic databases (PubMed, Embase and Web of Science Core Collection), grey literature and citation tracking. The time frame will be from 1 January 2000 to 31 December 2021. Only indices or assessment tools that are globally applicable and provide measurable indicators of health governance will be eligible. A qualitative content analysis will follow the proposed data extraction form to explicate and compare each eligible index or assessment tool. An analysis based on a proposed preliminary evaluation framework will identify the advantages and gaps and summarise the lessons learnt. This scoping review will also discuss the feasibility and necessity of developing a new global health governance index or consensus framework to inform future research and practices.

**Ethics and dissemination** This scoping review does not require ethics approval. Dissemination will include a peer-review article, policy briefs and conference presentations. This protocol has been registered in the Open Science Framework (osf.io/y93mj).

## STRENGTHS AND LIMITATIONS OF THIS STUDY

⇒ This scoping review will be a prior assessment in establishing a new index or consensus framework for assessing global health governance for the post-COVID-19 era.
⇒ This scoping review will differ from the existing reviews by incorporating governance for a wide range of health objectives and broadening geographic coverage with a global lens.
⇒ The literature to be reviewed will include research articles and indices or assessment tools used by organisations, with theoretical and practical implications for assessing health governance.
⇒ Pilot tests in searching and study selection and consultation with multiple librarians were conducted for the protocol development.
⇒ With the topic being broad and interdisciplinary, the precision of the search strategy might be constrained.

governance is described as a series of collective actions and decision-making procedures with diverse actors and organisations without formal control mechanisms.[1] Governance emphasises governing with and through networks between public, private and voluntary sectors.[2] It is one of the blocks in the widely-used health systems framework formulated by the WHO.[3] Given the globalised health issues, health governance in each sovereignty has been closely linked. From the pandemic of SARS to COVID-19, repeating global health crises have alerted the need for global health solidarity efforts.[4] However, there is still a lack of a solid governance framework under 'international anarchy',[5–7] although United Nations' 2030 Sustainable Development Goals have set up goals to promote global health outcomes.

Indeed, existing indices or assessment tools in global health tend to focus on health outcomes instead of the governance elements

## INTRODUCTION
### Rationale

The health governance of countries shapes global health governance. In a broad sense,

attributed to these outcomes (see Appendix A in the online supplemental material 1). Even within health governance, multiple parallel overlapping frameworks, assessment tools and indices for theoretical or practical purposes have created complexities. Besides, 85% of global health organisations have their headquarters in Europe or North America; more than 80% of the global health leaders come from high-income countries.[8] Therefore, most global health indices or assessment tools and indicators have been produced from high-income countries' perspectives, failing to reflect the other populations. Due to economic constraints and low logistic capacity, health statistics in developing countries are with varying standards and difficult-to-assess accuracy.[9] Thus, global health indicators' validity, utility and representativeness in developing countries are questionable.[10]

The underlying standpoint of this scoping review is that with the deeply rooted notions of sovereignty, global health governance has to be anchored around the health governance of countries. A starting point might be a consensus framework or a new, integrated index on health governance across countries globally. Thus, scoping the existing indices and assessment tools will lay a practical basis for developing an index or consensus framework to benchmark global health governance for equity and solidarity.

### Objectives

This scoping review aims to review the existing multi-country indices and assessment tools applied globally with measurable indicators assessing health governance; summarise their differences and commons; identify the lessons learnt through analysis of their advantages and gaps and assess the feasibility and necessity to establish a new index or consensus framework for assessing global health governance.

As global health governance is an emerging, multidisciplinary field, a scoping review is a more appropriate tool to 'assess and understand the extent of the knowledge and identify, map, report or discuss the characteristics or concepts'.[11] By contrast, systematic reviews aiming to 'answer a clinically meaningful question or provide evidence to inform practice'[12] or literature reviews with less systematic, transparent and reproducible methods will not meet the objectives above.

### Eligible literature

Only indices or assessment tools that are globally applicable and provide measurable indicators of health governance will be eligible. Indices and assessment tools are both tools for evaluation with measurable indicators. In practice, 'index' is often an external evaluation tool resulting in scores or rankings, while 'assessment tool' often refers to guidance or checklist for benchmarked standards (it might be called 'self-assessment tool' in some cases). Regarding 'health', as the One Health approach has attracted increasing attention but faced challenges in operationalisation within global health governance,[13] this scoping review will include indices or assessment tools related to human, animal and environmental health.

International institutions, universities and think tanks might have established the majority of the potentially eligible literature, such as the Global Health Security Index, International Health Regulations Monitoring & Evaluation Framework and the Ocean Health Index. Other potentially eligible literature can also be found in bibliographic databases, such as the 'health development governance index'.[10] In the health sector, the authors could only find indices or assessment tools to evaluate national or subnational governance, although the assessment results might be comparable across countries under international coordination. Therefore, the authors posit that the assessment of transnational, multinational, international or global health governance might be rare. However, the authors will include the latter pieces of literature if there are any.

This scoping review excludes assessment frameworks without measurable indicators for the following reasons. First, there have been scoping reviews, systematic reviews or review protocols covering health governance frameworks in the health system,[14–16] health emergencies or health security[17] or both,[18 19] while few of them pragmatically concentrate on indices or assessment tools. Second, most health governance frameworks have not been applied in practice, and there is a lack of real-world evidence to validate the efficacy of these frameworks. Pyone *et al* found that within 16 frameworks for assessing governance in the health system, only five were applied in empirical research.[16] Mikkelsen-Lopez and her colleagues also point out that the lack of empirical work might result from unrealistic indicators and overly complicated framework design.[20]

This scoping review excludes indices or assessment tools designed to be applied in a particular country or region. Some reviews have included indices or assessment tools applied in regions like Europe as part of eligible literature.[21–23] Moreover, considering the objectives of this scoping review, including indices or assessment tools applied in particular countries or regions will weaken the global generalisability. In addition, since the concept of 'governance' in this scoping review involves diverse actors and organisations, governance of only one type of organisation (eg, hospital or enterprise) does not fit this research's scope.

### Related published/ongoing reviews

The authors did not identify any published or ongoing systematic reviews or scoping reviews on the topic through a preliminary search in Google Scholar, PROSPERO, Joanna Briggs Institute (JBI) Evidence Synthesis, Figshare, Open Science Framework and Research Gate (see Appendix B in the online supplemental material 1 for the methods of the preliminary search). Some eligible indices or assessment tools included in similar reviews[23] will be included and analysed in this scoping review, although their objectives and analytical methods differ from those of this scoping review.

**Table 1** Search terms

| Key concepts | Health | Governance | Assess | Measuring tools | Global |
|---|---|---|---|---|---|
| Search terms | health | 1. governance<br>2. leadership<br>3. accountability<br>4. stewardship<br>5. transparency<br>6. policy development/formulation<br>7. strategic vision/direction<br>8. partnership<br>9. participation<br>10. involvement<br>11. consensus | 1. evaluate<br>2. monitor<br>3. measure<br>4. assess | 1. indicator<br>2. score<br>3. index | 1. global<br>2. international<br>3. world<br>4. multi-country |

Specifically, this scoping review will differ from the existing reviews by incorporating governance for a wide range of health objectives, such as health system strengthening (including universal health coverage) and health security (including public health emergency preparedness), broadening the geographic coverage with a global lens, and focusing on indices or assessment tools in practice to inform decision-making for future assessment of global health governance.

## METHODS

This scoping review protocol follows Arksey and O'Malley's methodological framework,[24] the JBI guidelines[25] and the Preferred Reporting Items for Systematic Reviews and Meta-analyses (PRISMA) methodology for scoping reviews.[11 26] The reviewers also refer to systematic review methods (eg, search strategy and reporting) that might assist the transparency and rigorousness of this scoping review.[27–31]

This protocol has been registered in the Open Science Framework (osf.io/y93mj). The searches were conducted in each proposed information source on 3 April 2022. The following research and writing will start in June 2022 and last 2–3 months. The final scoping review will report important protocol amendments and their rationales.

### Research questions

Following the objectives of this scoping review, the primary research question will guide the study: what indices or assessment tools are designed to assess health governance across multiple countries? Besides, two additional research questions are based on the primary question. First, what are their differences and commonalities? Second, what are the lessons learnt to inform the future global health governance index or consensus framework development?

### Identifying relevant studies
#### Electronic searches

The search strategy will locate both publications in bibliographic databases and grey literature and adapt for each included information source. Given that the term 'health governance' only became common in the published literature around 2000, the search will be filtered by the publication dates between 1 January 2000 and 31 December 2021. The Peer Review of Electronic Search Strategies checklist has been used for the proposed full search strategy.[31]

Our search terms come from the following sources: concepts related to research questions, MeSH (Medical Subject Headings) and Emtree databases and completed

**Table 2** Eligibility criteria: SOCT framework

| | Inclusion criteria | Exclusion criteria |
|---|---|---|
| Subjects | Indices or assessment tools on human, animal and/or environmental health governance with measurable indicators | Assessment frameworks, conceptual frameworks or narrative assessments without measurement; on topics irrelevant to health |
| Objectives | Describing the indices or assessment tools (including indicators or scoring system) | Only criticising, mentioning and analysing the indices or assessment tools while not aiming to yield assessment results for health governance |
| Coverage | Can be applied in multiple countries at the global level | Applied or can only be applied within one country, one region or one type of specific organisations or individuals (eg, hospital, enterprise); only appearing as a case study without further generalisation |
| Type of sources | Reports, documents, peer-reviewed publications, websites | Commentaries, editorials, reviews, blogs, letters, conference abstracts, protocols |

SOCT, Subjects, Objectives, Coverage, Type of sources.

**Table 3** Draft data extraction form

| Extraction category | Description | | Data type |
|---|---|---|---|
| Name | Full name of the index or assessment tool | | Unstructured text |
| Developer | Author or agency that developed the index or assessment tool | | Unstructured text |
| Reference | The reference information of the index or assessment tool | | Unstructured text |
| Time coverage | First publication year | | Numerical data |
| | Publication frequency | Number | Numerical data |
| | | Annual, biennial, quarterly, monthly, etc. | Categorical data |
| | The coverage of years the index or assessment tool being used | | Numerical data |
| Operation, if applicable | Roles and coordination among sponsor, funder, manager or other stakeholders | | Unstructured text |
| Domain | Human health, animal health, environmental health, etc. | | Categorical data |
| Issues to address | The health issues to address, for example, health system strengthening, health security or health data | | Categorical data |
| Objectives | The purpose of index or assessment tool creation; the assessed subjects | | Unstructured text |
| Geographic coverage | Number of countries assessed | | Numerical data |
| | The geographic regions of countries assessed, for example, Asia, Africa, Europe, North America, South America or global | | Categorical data |
| Implementation level | The implementation level that the index or assessment tool was designed to assess, for example, global, transnational, regional, national, subnational or local level | | Categorical data |
| Dimensions | The indicator dimensions (not the specific indicators) of assessment content, for example, leadership, accountability, transparency and policy development | | Categorical data |
| Indicators | The indicators measuring health governance | | Unstructured text |
| Theory or logic, if applicable | The theory or logic based to develop the index or assessment tool | | Unstructured text |
| Methods of index or assessment tool development | Methods of design and development of the index or assessment tool, for example, Delphi, review of literature or modelling | | Categorical data |
| Methods of data collection | The approach used to obtain information necessary for the assessment, for example, questionnaire, checklist, interview or secondary data collection | | Categorical data |
| Methods of yielding results | Methods of yielding assessment results, for example, qualitative, quantitative or mixed methods and the corresponding specific methods | | Categorical data |
| Types of assessment results (if there are any open ones) | Types of results present the assessment results, for example, scores, rankings and ratings | | Categorical data |
| Validity and reliability, if applicable | Description of the validation process or reliability check of the assessment | | Unstructured text |

and ongoing related systematic reviews and scoping reviews. Using table 1, the authors join all terms within each concept with OR and join each concept together using AND.

The authors will search the following bibliographic databases: PubMed, Embase and Web of Science Core Collection. Appendix C in the online supplemental material 1 presents a full search strategy for each electronic database.

Given that some indices or assessment tools might not be commercially or academically published, grey literature will be an essential source of information in this scoping review. Google will be searched using a decustomised mode. Other search tools will include WHO Institutional Repository for Information Sharing (IRIS). In addition, experts in global health will be consulted to explore additional literature sources.

### Citation tracking
As the meaning of "governance" in this scoping review might not be apparent in the existing indices or assessment tools, citation tracking will be used to identify relevant articles. One approach is backward snowballing (reference searching) through reviews or literature

**Table 4** Preliminary evaluation framework

| Criteria | Description |
|---|---|
| Indicator completeness | The extent to which the indicator system is complete and operationalised in the following ways (including but not limited to):<br>1. The indicators can be assigned a direct value without following implicit indicators or questions;<br>2. The indicators are predefined and organised, not being example indicators. |
| Clarity of measurement parameters | The extent to which the methods for measurement of the indicators, actions, or structures are stated |
| Being evidence-based | The extent to which the observational or experimental evidence is provided for assigning value to the indicators |
| Feasibility | The extent to which the index or assessment tool could be applied in multi-country settings in the following ways (including but not limited to):<br>1. It is inclusive of disparities of countries, with universal or flexible indicators and available data;<br>2. A management structure or accountable entity has been or is to be set for the long-term operation of the index or assessment tool. |
| Utility | The extent to which the index or assessment tool supports decisions related to improvement (aiming at internal audiences) or accountability (aiming at external stakeholders), and policy advocacy or other functions. |
| *Sustainability* | The extent to which the index or assessment tool could be applied continuously in the following ways (including but not limited to):<br>1. It has a long-term operating plan, or it has been applied for multiple years;<br>2. It accommodates changes in the health issues or other conditions;<br>3. It has predictable long-term technical, managerial and financing support for daily functioning. |

citing a potentially eligible index or assessment tool. For example, the scoping review by Chiossi *et al* might have included some potentially eligible literature for this scoping review.[23] Another approach is forward snowballing (cited by searching) through eligible literature. Citation tracking in the related field of literature can support us in finding additional indices and assessment tools.

### Selection of eligible studies

The literature that meets all the inclusion criteria will be included, while literature that meets any one of the exclusion criteria will be excluded. Table 2 presents the eligibility criteria, following the Subjects, Objectives, Coverage, Type of sources framework developed by the authors. Appendix D in the online supplemental material 1 presents detailed eligibility criteria to assist the reviewers' decision in study selection.

All literature searched through bibliographic databases will be uploaded to Covidence, which will identify and remove duplications. Based on the eligibility criteria, two independent reviewers will screen the titles and abstracts (and full texts if no clues are helping to judge the eligibility) and then assess the full texts in detail to select the literature. However, for Google and WHO IRIS, another two reviewers will decustomise the searching, export the results for each search string to Excel, screen the titles and abstracts, summaries or introductions if applicable, and then assess the full texts in detail separately. Literature obtained from citation tracking will be selected after

the selection process of literature obtained from electronic searches.

A pilot test with randomly selected 50 samples will be conducted. The reviewers will meet to discuss discrepancies and modify the eligibility criteria and elaboration document. The screening will only start when 75% agreement is achieved.[25]

The reasons for any exclusion following the full-text review will be recorded. The reviewers will resolve disagreements through discussions throughout the selection process. A third reviewer will make the final decision if the two paired reviewers cannot resolve the disagreement.

The search results and the study selection process will be reported in the final scoping review and presented in a PRISMA extension for scoping review flow diagram.[26] All data will be recorded and exported into Excel form after the whole process ends.

### Data extraction

Two reviewers will extract data from the eligible literature independently using a tailored data extraction tool developed by the authors (table 3). If discrepancies occur during the data extraction process, the two reviewers will discuss to reach a common decision. If there is an unsolved disagreement, a third reviewer will make the final decision. There will be a pilot test to ensure consistency among the reviewers.

The authors might modify the draft data extraction form during data extraction. The scoping review will detail the modifications compared with this protocol.

## Data presentation and analysis

A qualitative content analysis will follow the data extraction form to explicate further and compare each index or assessment tool.

Tables and figures will present the extracted data for each extraction category, followed by detailed descriptive analyses. An overview table will show the basic information of each eligible literature, including the name, developer and references. Then, numerical or categorical data will be calculated on counts and proportions. For instance, there might be N (p%) articles using Delphi approaches to develop the indices and assessment tools. Such statistics will help grasp an overview of the characteristics of the eligible literature. For unstructured texts, a qualitative data analysis software will be used for coding. The contents related to governance will be particularly coded. However, the data of some extraction categories with unstructured texts could probably transfer to numerical or categorical data. For example, theory or logic might be further categorised by disciplines.

To better identify the advantages and gaps and summarise the lessons learnt, there will be an analysis based on the proposed preliminary evaluation framework (table 4) after the data presentation. This framework is amended from Haeberer *et al*'s framework[22] according to the topic of this scoping review and the contents relying on the authors' subjective judgement were cut. The purpose of this framework is not to set criteria for the indices or assessment tools. Instead, it is simply to guide a further deep discussion based on the descriptive data.

Following the analysis above, this scoping review will discuss the feasibility and necessity of developing a new global health governance index or consensus framework. The feasibility evaluation in table 4 will facilitate the feasibility analysis at this stage, and the gaps identified above will assist the necessity analysis. Therefore, the study will inform future research and practices in assessing global health governance.

## Patient and public involvement

Patients and the public will not be involved in this scoping review.

## ETHICS AND DISSEMINATION

The analytical results will inform various stakeholders, including researchers, public health agencies, governments, global health organisations and other health governance actors. Dissemination of this scoping review will include publication in a peer-reviewed scientific journal, policy briefs and conference presentations. Ethics approval is not required as the data are available publicly.

**Acknowledgements** The authors are grateful to two librarians, Muriel Leclerc from the University of Geneva and Yi Ren from Tsinghua University, for their guidance on the search strategy. Also, thank Marhaba Abdurëhim, Junpeng Wang, and Yunke Gu for doing the initial search to check the availability of the potential indices or assessment tools.

**Contributors** All authors contributed to the study's design, drafted the manuscript, provided feedback and approved the final manuscript. KT provided feedback in principle, oversaw revisions and refined the manuscript. KT will also be the guarantor of the review. AH developed the search strategy, eligibility criteria, and data extraction tool and drafted and edited the protocol. YL structured the protocol, drafted the methods, refined the search strategy and eligibility criteria in detail, and contributed to the pilot tests. LZ drafted the introduction session and the supplemental material, refined the search strategy and the manuscript, and contributed to the pilot tests. JD developed the initial search strategy and eligibility criteria and drafted the methods. QH led the initial search to check the availability of the potential indices or assessment tools.

**Funding** This work is supported by the Research Fund of Vanke School of Public Health, Tsinghua University (Number 2021ZZ004) and Spring Breeze Fund, Tsinghua University (Number 20203080035). The funders are not involved in the research. They will also have no input on interpreting or publishing the study results.

**Competing interests** None declared.

**Patient and public involvement** Patients and/or the public were not involved in the design, or conduct, or reporting, or dissemination plans of this research.

**Patient consent for publication** Not applicable.

**Provenance and peer review** Not commissioned; externally peer reviewed.

**ORCID iDs**
Yuling Lin http://orcid.org/0000-0002-3834-4111
Kun Tang http://orcid.org/0000-0002-5444-186X

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
