## [Reviewer comments · BMJ Open]

ARTICLE DETAILS

TITLE (PROVISIONAL)	Assessing Health Governance Across Countries: A Scoping Review Protocol on Indices and Assessment Tools Applied Globally
AUTHORS	Huang, Aidan; Lin, Yuling; Zhang, Liyuan; Dong, Jingwen; He, Qiwei; Tang, Kun

VERSION 1 – REVIEW

REVIEWER	Tediosi, Fabrizio Schweizerisches Tropen- und Public Health-Institut
REVIEW RETURNED	12-May-2022

GENERAL COMMENTS	I would only suggest to revise the section "Data analysis and presentation" as it does not explain clearly enough how the analysis will be done. The sentences below do not allow to get how the analysis will be done "NVivo will be used in the coding process to analyse the specific indicators, and the role of "governance" will be particularly specified. This scoping review will also dialogue with existing critical analysis on the index and assessment tools eligible for this study. " I would even recommend to develop a sort of "framework of analysis" that will be applied in the review.
---

REVIEWER	Graeden, Ellie Georgetown University, Global Health Science and Security
REVIEW RETURNED	23-May-2022

GENERAL COMMENTS	General comment: The authors do not address the concept of One Health or otherwise speak to the impact of animal health on human health anywhere in the paper, but consistently use the term "health" in a general context. I would recommend inclusion of One Health for this analysis of indices. If only "human health" is to be addressed, this should be addressed up front. At a minimum, if One Health and zoonotic indices are not intended for inclusion, they should be listed as an exclusion criteria. General comment: Given the goal of this effort, I would suggest inclusion of the Joint External Evaluation and SPAR (Self-Assessment Annual Reports) tools in addition to the IHR. While designed to be used by individual countries, both are based on a global framework (the IHR) and are widely used tools in evaluating global health. Line notes: Abstract, Line 6 - Would suggest re-framing or clarifying the basis for the broad statement around the "failure of global health efforts".
--

	This is not supported by the literature cited, nor is it aligned with the stated goal of the research effort to evaluate indices and frameworks already in place that inform and measure progress forward global health implementation. Line 7 - “solidary” does not seem to be the correct word here. Did you intend “solidarity”? Page 5, Line 19 - “Nuclear Threat Initiative”, not “Threat Initiative” Page 5, line 33: This first question is listed as a research question, but is previously phrased as inclusion criteria. It should be one, not both. I would suggest that it remain as inclusion criteria; if a related question is a topic of the research, the research question should be clarified. Page 9 Table: For each category, it would be helpful to add a definition of either “unstructured text” or, where appropriate, define the structured terms to be used as metadata. Domain, geographic coverage, time coverage, implementation level, dimensions, and methods should, at a minimum, be structured with at least a proposed list of metadata to be applied. (“Methods” is the only category for which the specific structured categories are currently defined.) Appendix A page 14, line 40: I would not include the term “leadership” to be a synonym of “governance” in this context. Leadership is typically defined as a personal characteristic, not one of governments or governance frameworks, as are being evaluated in this study. If it is to be included, I would want to see a more robust explanation and validation of its inclusion as a search term. Appendix A, Table A: Please clarify the column headers in this table. The content and labeling is unclear.
--	---

VERSION 1 – AUTHOR RESPONSE

Response to reviewers’ comments:

Reviewer: 1

Mr. Fabrizio Tediosi, Schweizerisches Tropen- und Public Health-Institut

Comment: I would only suggest to revise the section “Data analysis and presentation” as it does not explain clearly enough how the analysis will be done. The sentences below do not allow to get how the analysis will be done “NVivo will be used in the coding process to analyse the specific indicators, and the role of “governance” will be particularly specified. This scoping review will also dialogue with existing critical analysis on the index and assessment tools eligible for this study. “

I would even recommend to develop a sort of “framework of analysis” that will be applied in the review.

Response: Thank you for the comment and recommendation. The analysis plan has been restructured, covering stages of data presentation, descriptive analysis and further discussion with frameworks more logical than the first manuscript (see Page 8-9). Thanks to Dr. Ellie Graeden’s comments on data extraction, we have accordingly refined the data presentation and analysis plan to make it more detailed and explicit.

Reviewer: 2

Dr. Ellie Graeden, Georgetown University

General comment: The authors do not address the concept of One Health or otherwise speak to the impact of animal health on human health anywhere in the paper, but consistently use the term “health” in a general context. I would recommend inclusion of One Health for this analysis of indices. If only “human health” is to be addressed, this should be addressed up front. At a minimum, if One Health and zoonotic indices are not intended for inclusion, they should be listed as an exclusion criteria.

Response:

Thank you so much for this comment. We agree that One Health is an important concept to clarify, and we have included One Health in the analysis. Therefore, we have made the following edits in the “Eligible literature” section:

- “Regarding “health”, as the One Health approach has attracted increasing attention but faced challenges in operationalisation within global health governance(13), this scoping review will include indices or assessment tools related to human, animal and environmental health.” (See Page 3, Line 57-59).

In Table 2, we have added the inclusion criterion: “Indexes or assessment tools on human, animal, and/or environmental health governance with measurable indicators”. In addition, we add a potentially eligible index (the Ocean Health Index) as an example on Page 5, Line 5.

General comment: Given the goal of this effort, I would suggest inclusion of the Joint External Evaluation and SPAR (Self-Assessment Annual Reports) tools in addition to the IHR. While designed to be used by individual countries, both are based on a global framework (the IHR) and are widely used tools in evaluating global health.

Response: Thank you for the suggestion. We will further screen the JEE and SPAR. Indeed, we had considered them as potentially eligible literature. Specifically, the “International Health Regulations (IHR) Monitoring & Evaluation Framework” mentioned on Page 5, Line 4-5 has four components: (1) States Parties self-assessment annual reporting (SPAR), (2) After action reviews (AAR), (3) Simulation Exercises (SimEx), and (4) Voluntary External Evaluations. The WHO also “recommends using the Joint External Evaluation (JEE) Tool for the voluntary external evaluation process”.

Line notes:

Abstract, Line 6 - Would suggest re-framing or clarifying the basis for the broad statement around the “failure of global health efforts”. This is not supported by the literature cited, nor is it aligned with the stated goal of the research effort to evaluate indices and frameworks already in place that inform and measure progress forward global health implementation.

Response: Many thanks for this comment. As suggested, we have summarised the literature cited and identified the current problems in global health indexes and assessment tools. We edited the first sentence in Abstract as follows:

- “Most global health indices or assessment tools focus on health outcomes rather than governance, and they have been developed primarily from the perspective of high-income countries.”

Accordingly, we’ve also refined the first half of the sentence on Page 4, Line 15-16 as follows:

- “However, there is still a lack of a solid governance framework under “international anarchy”...”

Line 7 - “solidary” does not seem to be the correct word here. Did you intend “solidarity”?

Response: Thank you for the comment. We have used the phrase “global health governance for equity and solidarity” on Page 3, Line 8 and corresponding wordings in Page 4, Line 35.

Page 5, Line 19 - “Nuclear Threat Initiative”, not “Threat Initiative”

Response: Many thanks for the comment. Considering the modification on One Health, we add potentially eligible literature (the Ocean Health Index) in this sentence. To keep the sentence succinct, we have deleted the affiliation information. Still, we have checked whether there are other similar mistakes and made corrections.

Page 5, line 33: This first question is listed as a research question, but is previously phrased as inclusion criteria. It should be one, not both. I would suggest that it remain as inclusion criteria; if a related question is a topic of the research, the research question should be clarified.

Response: Thank you for your advice. We have simplified and rephrased accordingly in the manuscript. Hereon we try to clarify that the first question on Page 6, Line 6, “what indices or assessment tools are designed to assess health governance across multiple countries?” is our primary question which guides and directs the development of the specific inclusion criteria for the scoping review.

Page 9 Table: For each category, it would be helpful to add a definition of either “unstructured text” or, where appropriate, define the structured terms to be used as metadata. Domain, geographic coverage, time coverage, implementation level, dimensions, and methods should, at a minimum, be structured with at least a proposed list of metadata to be applied. (“Methods” is the only category for which the specific structured categories are currently defined.)

Response: Thank you for the suggestion. On Page 8, we have refined Table 3 and predefined the data types (including unstructured text, categorical data and numerical data). Specifically, data like domain, issues to address, geographic coverage (the geographic regions of countries assessed), and implementation level could be grouped into categorical data. We have listed some examples of potential categories, whereas the final categories have to be induced at the review stage. This comment also reminded us to rethink the contents of Table 3 to guide our data extraction better. We have specified the data to be collected for time coverage and methods and added “issues to address” to distinguish it from “domain” and “indicators” to better analyse the indicators in the following stages. However, the draft data extraction table was only developed and piloted at the protocol stage according to our preliminary search results, and it may be further refined at the review stage through an iterative process.

Appendix A page 14, line 40: I would not include the term “leadership” to be a synonym of “governance” in this context. Leadership is typically defined as a personal characteristic, not one of governments or governance frameworks, as are being evaluated in this study. If it is to be included, I would want to see a more robust explanation and validation of its inclusion as a search term.

Response: Thank you for raising this issue. We have clarified our explanation for including the term “leadership” as our search term in Appendix A, Page 14, Line 3-12. We admit that “leadership” has multiple meanings, and there might be noise if we use it as a search term. However, given the link between “governance” and “leadership” embedded in current literature, we insist on including it to scope the literature assessing “leadership”. Albeit there might be a situation where few existing indexes or assessment tools assess “leadership”, it would be an interesting finding as “leadership” is commonly used to indicate governance.

Appendix A, Table A: Please clarify the column headers in this table. The content and labeling is unclear.

Response: Thank you so much for the comment. We have rephrased and tried to elaborate on Appendix A, Page 14, Table A, and refined the table's contents and labelling (see Appendix A, Page 13, Line 29-36). We've also double-checked the footnotes to make the selection of search terms more transparent.

Reviewer: 1

Competing interests of Reviewer: None

Reviewer: 2

Competing interests of Reviewer: None

Besides, inspired by the reviewers' comments, we have also made the following modifications:

- Use "indices" as the plural form of "index", which stands for "a standard by which the level of something can be judged or measured. In comparison, "indexes" rather refers to "an alphabetical list of names, subjects etc at the back of a book, or "a set of cards or a database containing information, usually arranged in alphabetical order and used especially in a library". (Source: Longman Dictionary of Contemporary English)
- Change the font size of the texts to ensure they fit with the line numbers.

Other modifications this time only aimed to refine the wordings and formats without changing the essential meaning of the original expression.

VERSION 2 – REVIEW

REVIEWER	Tediosi, Fabrizio Schweizerisches Tropen- und Public Health-Institut
REVIEW RETURNED	27-Jun-2022
GENERAL COMMENTS	The authors addressed the comments of the first review reasonably.
REVIEWER	Graeden, Ellie Georgetown University, Global Health Science and Security
REVIEW RETURNED	25-Jun-2022
GENERAL COMMENTS	No additional comments or changes.